# Monitoring the Dilution of Buffer Solutions with Different pH Values above and below Physiological pH in Very Small Volumes

**DOI:** 10.3390/s24175751

**Published:** 2024-09-04

**Authors:** Vinayak J. Bhat, Daniel Blaschke, Sahitya V. Vegesna, Sindy Burgold-Voigt, Elke Müller, Ralf Ehricht, Heidemarie Schmidt

**Affiliations:** 1Leibniz Institute of Photonic Technology, 07745 Jena, Germany; 2Institute of Solid State Physics, Friedrich Schiller University Jena, 07743 Jena, Germany; 3InfectoGnostics Research Campus, 07743 Jena, Germany; 4Institute of Physical Chemistry, Friedrich Schiller University Jena, 07743 Jena, Germany

**Keywords:** PolCarr^®^ impedance chip, dilution, equivalent circuit modeling, impedance measurement, impedance spectroscopy, buffers

## Abstract

The accurate determination of the post-dilution concentration of biological buffers is essential for retaining the necessary properties and effectiveness of the buffer to maintain stable cellular environments and optimal conditions for biochemical reactions. In this work, we introduce a silicon-based impedance chip, which offers a rapid and reagent-free approach for monitoring the buffer concentrations after dilution with deionized (DI) water. The impedance of the impedance chip is measured, and the impedance data are modeled using a multiparameter equivalent circuit model. We investigated six aqueous biological buffers with pH values above and below the physiological pH for most tissues (pH ~ 7.2–7.4) following dilution with DI water by factors of 2.0, 10.0, 20.0, 100.0, and 200.0. The impedance measurement is then performed for the frequency spectrum of 40 Hz to 1 MHz. From the interpretation of the impedance measurement using the multiparameter equivalent circuit model, we report a buffer-sensitive equivalent circuit parameter R_Au/Si_ of the silicon-based impedance chip showing a linear trend on a logarithmic scale with the buffer concentration change after dilution. The parameter R_Au/Si_ is independent of the buffer pH and the added volume. The results demonstrate the efficacy of the silicon-based impedance chip as a versatile tool for precise post-dilution concentration determination of diverse biologically relevant buffers. The presented impedance chip offers rapid, accurate, and reliable monitoring, making it highly suitable for integration into automated liquid-handling systems to enhance the efficiency and precision of biological and chemical processes.

## 1. Introduction

Biological buffers are essential for providing stable pH conditions, creating an optimal environment for cellular [1] and biochemical processes [2,3], maintaining the stability of cell membranes [4], and preserving the integrity of DNA and RNA molecules [5]. In biopharmaceutical manufacturing, buffers play an important role during upstream and downstream processing, including fermentation and purification [6,7]. Dilution of biological samples is crucial for adjusting concentrated stock buffers to a desirable working range and optimizing biological activity. However, a significant reduction in buffer concentration after dilution can diminish the buffer’s capacity to maintain optimal conditions, particularly in regulating pH. Buffers with a pH below the physiological pH, which is most effective in maintaining the stable pH of a biological system (normally 7.2–7.4 pH at room temperature [8,9]), can lose their effectiveness with overdilution, leading to a significant increase in pH and a reduced ability to prevent the media from becoming alkaline. Conversely, the same can be said for biological buffers with a pH above the physiological pH [10]. Therefore, monitoring buffer dilutions and knowing the post-dilution concentration is crucial for maintaining buffer integrity. Recent methods for continuous monitoring of buffer capacity include the use of optical sensors and real-time online titration methods. Mistlberger et al. [11] presented a dynamical optode in the optical sensor in their research to optimally detect and measure the buffer capacity. Paulsen et al. [12] used an acid–base titration technique in preparing the buffer suitable to calibrate and monitor the seawater pH measurements. The optical sensors technique uses the information from the interaction of the light with the buffer solution, either changes in the refractive index of the solution or changes in the emitted fluorescence intensity, which could be related to the buffer concentration after dilution [13,14]. Optical sensor techniques exhibit advantages like fast response and easy integrability but are limited by the specificity of the requirement of specific buffer components [15,16]. In the case of real-time online titration, continuous sampling of the test solution from the titration system is performed, and with the automated addition of titrant, a traditional titration is performed continuously [17,18]. The titration method is a well-established technique, but its accuracy is affected by the flow rate of the sample and titrant, and the method also requires high maintenance. Additionally, sensors that detect changes in electrical properties due to buffer concentration have the potential for performing continuous dilution concentration measurements. These sensors provide continuous data on buffer concentration, allowing for immediate adjustments to maintain optimal conditions, which is crucial in applications such as pharmaceutical production, food and beverage processing, and chemical manufacturing [19].

This work introduces the potential of a silicon impedance chip for monitoring and determining the dilution concentration of biological buffers. Silicon surfaces exhibit high sensitivity toward the properties of liquids they come in contact with, including polarity, surface tension, and ionic strength [20]. These properties influence the electrical characteristics of the silicon surface, such as conductivity and charge carrier mobility, which can be accurately assessed using impedance spectroscopy. Variations in buffer concentration lead to changes in impedance profiles, which enable the precise quantitative determination of concentration levels in the buffer. In our previous work, a silicon-based impedance chip was presented in determining the concentration of the aqueous phosphate buffer [21], with three different pH values below the physiological pH from 5.50 to 7.12. The impedance chip has already been utilized as a biochip, employing the same technique for applications such as bacterial viability testing [22] and bacterial cell counting [23]. In this report, we investigated the impedance response of six biological buffers on the silicon impedance chip. Three of the used buffers had a pH greater than the physiological pH, and the other three had a pH lower than the physiological pH. This selection of biological buffers with a wide spectrum allows us to evaluate the efficacy of the impedance chip with buffers of varying ionic strength. The proposed impedance chip has the potential advantage of overcoming the limitations of optical sensors and online titration techniques by leveraging the sensitive response of silicon surfaces to liquid properties. Furthermore, the impedance chip directly determines the concentration for a small volume of the buffer added without additional reagents, simplifying the process and reducing maintenance compared to other methods. Additionally, the proposed impedance chip holds promise for integration into automated liquid-handling systems (ALHSs) commonly used for liquid dispensing and mixing in both laboratory [24,25] and industrial settings [26,27]. By incorporating the chip directly into the dispenser’s flow path, real-time feedback on buffer concentration becomes possible. This feedback signal can be sent to the integrated controllers for real-time re-adjustment during the mixing process. The integration has the potential to increase throughput and further enhance the automation capabilities of ALHSs.

## 2. Materials and Methods

### 2.1. Buffer and Nutrient Media Preparations

For investigation, buffer and culture media preparations (Table 1) were used, which are indispensable in the processing of bacteriophage solutions. PBS buffer (phosphate-buffered saline, test solution 1, Table 1) was prepared by weighing the exact amounts of the chemicals, which are calculated according to the desired concentration of 137 mM NaCl, 2.7 mM KCl, 4.3 mM Na_2_HPO_4_·7H_2_O, and 1.4 mM KH_2_PO_4_ (Carl Roth GmbH, Karlsruhe, Germany). Demineralized water was added and mixed thoroughly to ensure that the chemicals dissolve completely. The pH of the PBS buffer was then adjusted to the desired value of pH = 7.2. The finished PBS buffer was then filled into sterile containers and sterilized by passing it through a Rapid-Flow bottle top filter (Thermo Fisher Scientific, Waltham, MA, USA). Luria–Bertani (LB medium, test solution 2, Table 1) was prepared from the following components, which were weighed out: Tryptone (Becton, Dickinson and Company, Sparks, NV, USA): 10.0 g/L; NaCl (Carl Roth GmbH, Karlsruhe, Germany): 10.0 g/L; and yeast extract (Becton, Dickinson and Company, Sparks, NV, USA): 5.0 g/L. Deionized water was made up to one liter, and the medium was autoclaved. Test solution 3 was prepared by filtering test solution 2 through a 0.2 µm cellulose acetate (CA) filter (Sartorius, Göttingen, Germany). A total of 0.1 M sodium bicarbonate buffer (test solution 5, Table 1) was prepared by diluting out of a 1 M stock solution (ThermoFisher (Kandel) GmbH, Kandel, Germany). The diluted buffer was transferred to an Amersham NAP-5 column (Cytiva, Marlborough, MA, USA), and the eluate was measured as test solution 4. Test solution 6 was prepared by processing solution 3 over a NAP-5 column and eluting with sodium bicarbonate buffer (solution 5, Table 1).

### 2.2. Silicon-Based Impedance Chip

A boron-implanted impedance chip (BG) was used in investigating the six buffer solutions for the post-dilution concentration measurement. Boron ions (B^+^) were implanted during the fabrication process into the phosphorous-doped silicon wafers. These phosphorous-doped silicon wafers are of 4-inch diameter with 525 µm thickness and serve as a substrate. The depth of the ion implantation is approximately 200 nm and establishes the p–n junction within the substrate. A gold (Au) ring electrode with inner and outer diameters of 5.7 mm and 7.8 mm was deposited onto the wafer. Subsequently, an unstructured gold bottom electrode is deposited using DC magnetron sputtering. The 4-inch wafer is then cut into 1 × 1 cm^2^ pieces.

The maximum amount of buffer solution that can be filled inside the ring electrode is limited by the height of the ring electrode and the hydrophobicity of the silicon surface. Cleaning treatment is performed on the impedance chip to maintain the desired hydrophobicity, which is determined by the contact angle of water on the silicon surface. Cleaning treatment involves the impedance chip being immersed in a Tergazyme bath for 1 h, followed by 1 h of a distilled water bath. Later, the impedance chip is treated with a 5% hydrofluoric acid (HF) solution for 30 s, followed by annealing at 150 °C for 20 min. Several impedance chips were used for cleaning, and those with the desired contact angle between 55 and 60 degrees were chosen. The water contact angle on the silicon surface between 55 and 60 degrees facilitates the introduction of a maximum volume of buffer solution up to 40 µL into the ring electrode. The contact angle measurement was performed using the Krüss Drop Shape Analyzer (DSA25E) (KRÜSS GmbH, Hamburg, Germany) measurement setup using the available Sessile Drop Method technique.

A photographic view of the 1 × 1 cm^2^ impedance chip wire-bonded to a TO-39 package is presented in Figure 1. Additionally, various liquid volumes of 20 µL, 26 µL, and 40 µL inside the ring electrode are also shown in Figure 1a, Figure 1b and Figure 1c, respectively. To ensure a uniform distribution of the electric field, the liquid within the ring electrode is consistently maintained in contact with the electrode. The height of the liquid for the initial addition of 20 µL inside the ring electrode is measured to be 1.24 mm (Figure 1d), for 26 µL, it is 1.55 mm (Figure 1e), and for 40 µL, it is 2.26 mm (Figure 1f). Recalculating the volume using the height of the liquid drop and the diameter of the inner ring electrode via the ellipsoidal volume equation results in an error margin of ±1 µL for the total volume inside the ring electrode. Given that the height of the liquid is significantly greater than the 150 nm thickness of the gold ring electrode, it is crucial to maintain the defined contact angle of the silicon surface to prevent overflow without losing the sensitivity of the silicon surface to the introduced liquid to its surface.

Frequency-dependent impedance measurements were performed utilizing an Agilent 4294A impedance analyzer (Agilent Technologies, Santa Clara, CA, USA), spanning frequencies from 40 Hz to 1 MHz. Both the empty impedance chip and the chip filled with buffer solution underwent analysis. The temperature remained constant at 21 °C throughout the measurements, with each impedance measurement consuming roughly 1 min to capture 100 data points across the frequency spectrum. The measurements were conducted under dark conditions.

## 3. Equivalent Circuit Modeling of Impedance Chips before and after Filling

Figure 2 demonstrates the equivalent circuit model with capacitance and resistance pairs, which is used to model the small changes in the impedance measurement of the filled impedance chip due to the change in volume, dilution, and pH. The proposed equivalent circuit is based on the geometry of the impedance chip, which is segmented into three blocks. The capacitance C_Si(n)/Au_ and resistance R_Si(n)/Au_ pair in the rightmost block represent the depletion region at the junction of the bottom gold contact and the phosphorus-doped silicon (n-Si) bulk. The depletion region formed at the p–n junction after the implantation of the boron(B^+^) ions is represented by the capacitor C_p_^+^_n_ and resistor R_p_^+^_n_. The leftmost block represents the depletion formed at the impedance chip surface between the gold ring electrode, filled buffer, and the silicon (Si(p^+^)) surface. The Schottky contact formed between the metal and semiconductor is represented by the parallel combination of a capacitor and a resistor. The top part of the leftmost circuit block includes a parallel combination of the capacitor C_Au/Si(p_^+^_)_, which represents the capacitance of the depletion region, and a resistor R_Au/Si(p_^+^_)_, which represents the potential leakage path between the gold ring electrode and the Si(p^+^) surface under the gold ring electrode. In the bottom part of the circuit block, there are two capacitance and resistance pairs. An electric double-layer formation between the gold electrode metal and the filled buffer liquid is represented by the parallel combination of C_Au/liq_, which represents the double-layer capacitance, and R_Au/liq_, which represents the charge transfer resistance at the liquid–gold ring electrode junction. The second pair C_j_ and R_j_ represent the space charge region between the native silicon dioxide and Si(p^+^) surface, and R_liq_ represents the bulk resistance of buffer liquid in contact with the Si(p^+^) surface. The series connection of two capacitance and resistance pairs of the bottom part of the circuit block gives the cumulative impedance of the interaction at the interface between the gold ring electrode, the buffer liquid, and the p-doped silicon (Si(p^+^)) surface.

During the impedance modeling, initially, the impedance data of the impedance chip with no filling are modeled. The initial modeling is performed by putting the calculated values of capacitors, such as the bottom contact capacitance C_Si(n)/Au_, p–n junction capacitance C_p_^+^_n_, and top contact capacitance C_Au/Si(p_^+^_)_, and the values are varied in a similar range during the modeling. The calculation of these capacitance values is from the known carrier concentrations N_A_ and N_D_, which are 400 × 10^15^ cm^−3^ and 2.5 × 10^15^ cm^−3^ of the implanted (B^+^) and doped (P^+^) ions, respectively. The permittivity of silicon is 11.7, and the area of the top electrode is 22.5 mm^2^. The calculation can be referred in our previous work [21]. For the empty chip, modeling the values of the liquid and silicon-surface-related parameters is kept high such that the cut-off frequency of the capacitance and resistance pair is less than 40 Hz and does not affect the modeling after 40 Hz. In the next step, modeling of the measured data of the impedance chip with 20 μL of buffer solution filled inside the ring electrode is performed by varying the parameters related to the liquid and silicon surface (leftmost block in Figure 2), such as C_Au/liq_, R_Au/liq_, C_j_, R_j_, and R_liq_. For the first addition of 20 μL inside the ring electrode, the bulk parameter is also adjusted to obtain the proper fit (center and rightmost block in Figure 2). In the next subsequent steps, as with the incremental additions of 2 μL until the total volume of 40 μL inside the ring electrode, the liquid and silicon-surface parameters, as well as the C_Au/Si(p_^+^_)_ and the resistor R_Au/Si(p_^+^_)_ pair, are varied, keeping other parameters unchanged for the modeling. The measured absolute impedance values and the modeled fitting for one of the buffer dilutions are shown in Appendix A. 

## 4. Results and Discussion

We performed the impedance measurement of the differently diluted test buffer solutions on different impedance chips, which are of the defined contact angle, by introducing up to 40 µL of the buffer test solution inside the defined ring electrode of the impedance chip. During the measurement, first the empty impedance chip without adding the test solution inside the ring electrode is measured, and then 20 µL of the buffer test solution inside the ring electrode is added. The added 20 µL test solution would require ~5 min to attain equilibrium with the silicon surface of the impedance chip. Impedance is measured after the equilibrium; subsequently, +2 µL is added until the total amount of buffer test solution is 40 µL inside the ring electrode, and impedance is measured for each time the additional +2 µL is added. The measured complex impedance is plotted on the Nyquist plot with the real part of the impedance on the x-axis and the imaginary part on the y-axis. During the behavioral analysis of the absolute values of the measurement, the complex impedance values of the empty impedance chip are subtracted from the obtained complex impedance values for the measurement with an added buffer test solution inside the ring electrode. This obtained effective impedance value after subtraction ensures the validation of the comparison of the different buffer test solutions on different impedance chips. The effective impedance values shown in Figure A2 therefore represent the changes in the impedance signal due to combined changes in the volume, pH, and buffer dilution concentration.

The effective impedance for the added 26 µL volume (Figure A2) inside the ring electrode exhibits increased complex impedance values compared to the DI water impedance (indicated by the blue solid line) for test buffers with higher concentrations. Upon dilution, the impedance values decrease and approach the DI water impedance, as anticipated. Buffer test solutions demonstrate a gradual reduction in complex impedance toward the DI water value with dilution. Notably, buffer test solution 5 (0.1 M NaHCO_3_, sky blue) shows significantly higher impedance values at a higher dilution factor, with the effective complex impedance decreasing at dilution factors 100 and 200 from the initial undiluted concentration. However, the complex impedance depicted in Figure A2 represents a combined effect influenced by the volume, pH, and dilution. Also, the spiral nature of the effective impedance data in the Nyquist plot shown may be attributed to the frequency-dependent interfacial impedance changes, especially at the interfaces between the buffer, silicon surface, and the gold ring electrode. This behavior is further influenced by the buffer’s ionic strength, pH, and viscosity. The capacitive behavior at different interfaces, combined with charge transfer resistance, leads to the observed distortions in the Nyquist plot, resulting in the spiral appearance. To distinguish and quantify these variations, parameters related to the liquid–silicon–ring electrode interface, as illustrated in the leftmost block of the equivalent circuit in Figure 2, are modeled. The modeling result parameters of the complex impedance of all the test buffer solutions are shown in Appendix A. From the obtained modeling parameters, we observe that out of the parameters related to the changes due to the interaction at the interface of buffer test solution–silicon and gold ring electrode, R_Au/Si_ shows a very low value of resistance for the higher concentration (lower dilution factor) of the buffer solution, and the resistance of the R_Au/Si_ parameter increases with the decrease in the concentration (higher dilution factor) of the buffer solution after dilution.

In Figure 3a, the selected three buffer test solutions (Buffer 1, 2, and 3) above the physiological pH and three buffer test solutions (Buffer 4, 5, and 6) selected below the physiological pH show the change in pH with the increasing dilution. The pH of the first three test buffers increases gradually with the dilution, with an offset for buffer test solution 2 at a dilution factor of 100 toward the DI water pH, whereas the other three buffers’ pH decreases slightly until a dilution factor of 10 and steeply toward the DI water pH with further dilution. The parameter R_Au/Si_ shows linear change with the dilution on the logarithmic scale, as illustrated in Figure 3b.

Notably, the R_Au/Si_ values for the buffer solution with pH values above physiological pH at initial undiluted concentration show a linear increase with the dilution up to 20 dilution factor and almost saturated R_Au/Si_ values for further dilution of the buffer. The buffer test solutions with pH below physiological pH at an undiluted solution concentration show an exponential increase in R_Au/Si_ values until the dilution factor of 100 and show slight variation or remain constant for further dilution. The independence of the R_Au/Si_ parameter from the volume is illustrated in Appendix B, Figure A1, where circuit parameter R_Au/Si_ is plotted against the volume added inside the ring electrode. For the initial added volumes, the R_Au/Si_ value shows slight variations. However, from 26 µL and for all subsequent volume additions, the R_Au/Si_ value remains constant. The parameter R_Au/Si_ depends on the ionic strength of the buffer. As the dilution increases, the ionic strength decreases, leading to a corresponding increase in R_Au/Si_. The saturation of R_Au/Si_ at higher dilutions indicates that the system’s electrical properties stabilize, resulting in minimal additional changes in impedance. This demonstrates that the impedance chip effectively monitors changes in ionic strength, independent of pH or volume, providing a reliable measure of buffer dilution. The remaining parameters C_j_, R_j_, C_Au/liq_, R_Au/liq_, and R_liq_ associated with the changes at the interface of buffer test solution, silicon and gold ring electrode, show the deviation due to the combined effect of pH change of the buffer, added volume inside the ring electrode, and the change in concentration of buffer after dilution. The values of each parameter against the dilution factor for all six buffers are plotted in Appendix A.

While impedance spectroscopy has been used by Chen et al. [28] to demonstrate online monitoring of pH using a tantalum pentoxide-based electrolyte-ion sensitive membrane-oxide-semiconductor (EIOS) pH sensor, which effectively provides the pH-sensitive capacitive parameter from a circuit fitting model for measured capacitance for the applied DC bias. Impedance techniques have also been extensively used to monitor cell profiles [29,30]. However, their application to quantitatively monitor buffer dilution concentration using a silicon-based impedance chip for applied AC bias is novel. The impedance chip quantifies changes in impedance as a function of buffer concentration and offers improved precision in real-time monitoring of the concentration of diluted buffers, which is advantageous for industrial applications. The impedance chip can be integrated into the different stages of the buffer mixing streamlines, like feedback systems or in the monitoring units of the buffer mixing. After mixing the concentrated buffer with the DI water in the mixer module, the diluted buffer flows through the integrated impedance chip. By monitoring the R_Au/Si_ parameter from the obtained impedance data, monitoring and controlling the buffer concentration within the desired specifications can be obtained. While in ALHS, the diluted buffer can be introduced through automated pipetting to the impedance chip, which is integrated into defined microwell plates connected to the monitoring unit, where the R_Au/Si_ parameter can be used to give feedback on the diluted concentration of the buffer.

## 5. Conclusions

This study presents a precise, real-time monitoring approach to measure the diluted buffer concentration using the silicon impedance chips with the ring top electrode. The primary finding is the ability of the impedance chip to offer quick and reagent-free measurement of post-dilution buffer concentration with a very small volume of the buffer, which potentially offers better accuracy and sensitivity in comparison with the existing monitoring technique. For the analysis, the impedance response of the six buffers, with pH values above and below the physiological pH, is measured at an undiluted concentration and after being diluted to factors of 2.0, 10.0, 20.0, 100.0, and 200.0. A circuit parameter, R_Au/Si_, from the developed equivalent circuit model effectively captures the impedance response of the post-diluted buffer concentration. The R_Au/Si_ parameter shows the linear dependency on a semi-logarithmic scale with the variation of buffer concentration, independent of pH, buffer type, and volume of the buffer. The impedance chip’s capability to detect concentration at small volumes minimizes the sample waste. Furthermore, using R_Au/Si_ of an impedance chip as an indicator of buffer concentration has an advantage for easy integration into the automation system for maintaining product quality and process consistency in buffer preparation with high sensitivity and increased efficiency in bioprocessing and other life science applications.

## Figures and Tables

**Figure 1 sensors-24-05751-f001:**
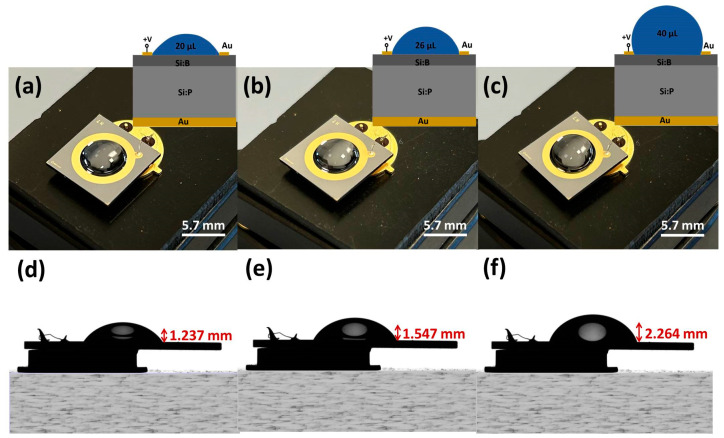
Photographed top view of the boron implanted p–n junction-based silicon impedance chip with the top gold ring electrode, showcasing the varied liquid volumes of (**a**) 20 μL, (**b**) 26 μL, and (**c**) 40 μL filled inside the ring electrode. The top gold electrode of the impedance chip is wire-bonded to the TO-39 socketing. A cross-section impedance chip schematic with the buffer inside the ring electrode is shown in the inset. Camera captured cross-sectional view (**d**–**f**) shows the height difference of the liquid drop within the top ring electrode of the impedance chip for the corresponding volume additions of (**d**) 20 μL, (**e**) 26 μL, and (**f**) 40 μL inside the ring electrode.

**Figure 2 sensors-24-05751-f002:**
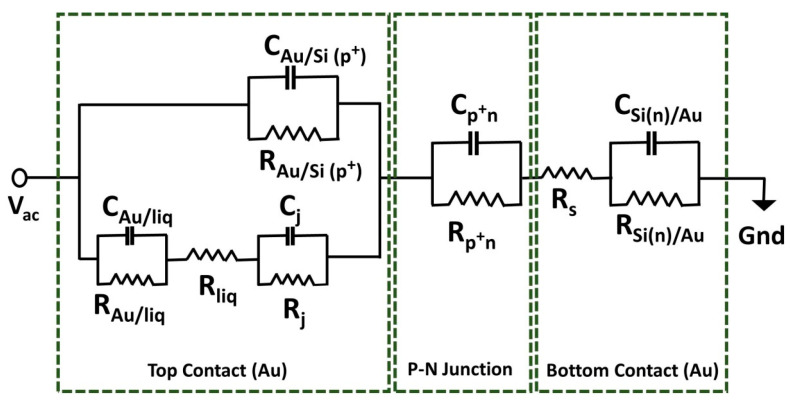
Illustration of the equivalent circuit model of the impedance chip to model the measured impedance of the empty and filled buffer liquid inside the ring electrode of boron-doped silicon impedance chips. The equivalent circuit is divided into three regions represented by rectangular green boxes. The rightmost box represents the Si(n) and bottom gold contact interface; the center rectangular box represents the p–n junction, and the leftmost rectangular box represents the top ring electrode, Si(p^+^), and added buffer liquid interface. The circuit parameters are modeled for both the empty and the first added volume of 20 μL inside the ring electrode. For the more added volume of the buffer liquid (22 μL until 40 μL) inside the ring electrode, the parameters in the leftmost box are modeled, keeping other parameters unchanged.

**Figure 3 sensors-24-05751-f003:**
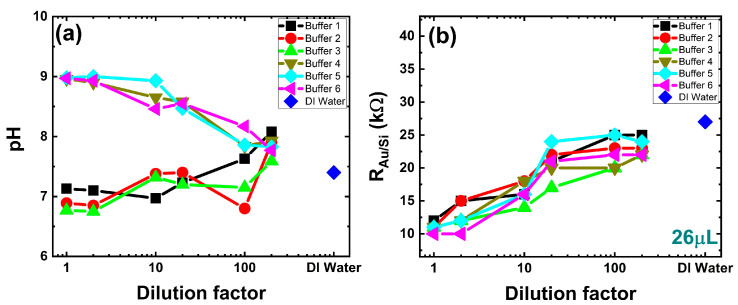
(**a**) Measured pH values of six buffers, obtained using a Schott CG 842 pH tester at 22 °C, plotted against the dilution factors of the buffers from the initial concentration, and (**b**) modeled equivalent circuit parameter R_Au/Si_ against the six diluted buffers with a dilution factor from 1.0 to 200.0 for the added buffer volume of 26 μL inside the ring electrode of the impedance chip. The R_Au/Si_ parameter shows a linear upward trend across all six buffers, regardless of their pH variations. For the pure DI water, the value of R_Au/Si_ is 27 kΩ and is represented by a diamond dot.

**Table 1 sensors-24-05751-t001:** Six different biological buffer solutions were used to investigate the post-dilution concentration using a silicon-based impedance chip. The experimental pH values determined using a Schott CG 842 pH tester at 22 °C for each buffer at an initial undiluted concentration are provided in the table.

Test Solution	Name	pH
1	PBS s.f.	7.12
2	LB	6.89
3	LB filtered/0.2 µm CA filter	6.77
4	0.1 M NaHCO_3_ + NAP-5	8.96
5	0.1 M NaHCO_3_	8.98
6	LB filtered/0.2 µm CA filter + NAP-5 + 0.1 M NaHCO_3_	8.97

## Data Availability

The data that support the findings of this study are available from the corresponding author, H.S., upon reasonable request.

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
