# Peer review of "Monitoring the Dilution of Buffer Solutions with Different pH Values above and below Physiological pH in Very Small Volumes"

_sensors, 2024, doi:10.3390/s24175751_

Round 1

Reviewer 1 Report

Comments and Suggestions for Authors

This paper presents a method based on silicon impedance chips to monitor the dilution of buffer solutions. It might be attractive for some readers. Some suggestions are as follows: 1) the presentation of Abstract should be re-organized. At the beginning of the paragraph, the authors should let readers know the novelty and the problems to be solved in this paper. 2) Too many spellings are not scientific, such as Na2HPO4*7H2O, g/l, B+,μl...... 3) There is a lack of comparison with other reports and methods in the part of Results and Discussion. 4) The format of the references should be unified.

Comments on the Quality of English Language

Proper but can be improved further

Reviewer 2 Report

Comments and Suggestions for Authors

The comments are included in the attached report

Best regards

Reviewer 3 Report

Comments and Suggestions for Authors

I received the paper entitled “Monitoring the dilution of buffer solutions with different pH values above and below physiological pH in very small volumes” for review and found it to contain interesting results and discussions of an important research topic within the field of liquid processing, biological samples testing, pharmaceuticals, food/drink industries, etc.

I have the following remarks that will make manuscript more valuable and provide more insightful details for prospect readers:

1- Correction suggestion in line 70: (…. liquid they come in contact with, including polarity…)

2- It will be good to state the meaning and range of values of the “physiological pH” in line 77.

3- From line 124 to line 132, please state if you used only one chip for testing all samples or several chips were used. If only one chip, how did you perform necessary cleaning after each test.

4- Try to use a clearer contrast of colours (or grey scale) in figure1 (d, e and f). The mentioned 3 figure subsets are mostly black.

5- Please state the origin/reference of the electric model presented in figure 2, or alternatively, if it’s your own model, then briefly justify its design (especially the parallelism of components in the left-most segment).

6- In the conclusion section; It will be valuable if the authors discussed/recommended how the chips will be used in industrial setups to obtain real-time measurements of pH. Will the Agilent impedance analyser be suitable in practice or will specific signal processing circuits be designed instead?

7- In the conclusion section; It will also be useful if the authors discussed the issue of liquid sampling in real-time ALHS, in other words, how will the samples be automatically introduced onto the chip for analysis?

Reviewer 4 Report

Comments and Suggestions for Authors

Line 113 (test solution 5, Table 1)

Line 115-116 test solution 4 

The information is presented adequately, but I think that figures A1.1 and A1.2 would be better inserted in the results and discussion text.

The presentation of their results leaves no room for doubt about the potential application of these small electrodes for pH monitoring, even in different solution concentrations and volumes.

Round 2

Reviewer 1 Report

Comments and Suggestions for Authors

Questions have been solved properly. This paper can be accepted.

Author Response

Thank you for your positive feedback and for acknowledging the efforts made in addressing the questions. We are grateful for your time and valuable comments, which have helped improve the quality of our manuscript.